# Hemodialysis Efficiency Predictor in End-Stage Kidney Disease Using Real-Time Heart Rate Variability

**DOI:** 10.3390/biomedicines12030474

**Published:** 2024-02-20

**Authors:** Sung Il Im, Ye Na Kim, Hyun Su Kim, Soo Jin Kim, Su Hyun Bae, Bong Joon Kim, Jung Ho Heo, Yeonsoon Jung, Hark Rim, Sung Pil Cho, Jung Hwan Park, Ho Sik Shin

**Affiliations:** 1Division of Cardiology, Department of Medicine, Kosin University Gospel Hospital, Kosin University College of Medicine, Busan 606-701, Republic of Korea; sungils8932@naver.com (S.I.I.); kim.hyunsu100@gmail.com (H.S.K.); circleabc@naver.com (S.J.K.); rodi0203@naver.com (S.H.B.); bongjoon81@hanmail.net (B.J.K.); duggymdc@gmail.com (J.H.H.); 2Renal Division, Department of Internal Medicine, Gospel Hospital, Kosin University College of Medicine, Busan 606-701, Republic of Korea; velvetrabbit21@hanmail.net (Y.N.K.); kidney@hanmail.net (Y.J.); rimhark@hanmail.net (H.R.); 3Transplantation Research Institute, Kosin University College of Medicine, Busan 606-701, Republic of Korea; 4MEZOO, Won Ju 26354, Republic of Korea; spcho@me-zoo.com (S.P.C.); jhpark@me-zoo.com (J.H.P.)

**Keywords:** predictors of hemodialysis, heart rate variability

## Abstract

Background: Autonomic dysfunction as a long-term complication may occur in end-stage kidney disease (ESKD) patients and can be diagnosed using heart rate variability (HRV) analyzed from electrocardiogram (ECG) recordings. There is limited data about HRV using real-time ECG to predict hemodialysis (HD) efficiency in patients with ESKD who are routinely doing HD in the real world. Methods: A total of 50 patients (62.1 ± 10.7 years) with ESKD underwent continuous real-time ECG monitoring (237.4 ± 15.3 min) during HD for HRV using remote monitoring system. Their electrolyte levels were checked before and after HD. We compared HRV according to electrolyte levels. Results: During the monitor, we checked the ECG and electrolyte levels simultaneously a total of 2374 times for all of the patients. Both time and frequency domain HRV were higher when the patients had lower K^+^ level (<0.5 mEq/L) and P^+^ level change (<2 mEq/L) before and after HD as compared to those with a higher K^+^ level (≥0.5 mEq/L) and P^+^ level change (≥2 mEq/L). Additionally, patients with lower K^+^ and P^+^ level change groups had higher incidences of arrhythmic events including atrial/ventricular premature complexes, despite no difference of mean heart rate (*p* < 0.001). Conclusions: Higher HRV was independently associated with a poorly controlled K^+^ and P^+^ level during HD in patients with ESKD. This is consistently evidenced by the independent association between higher HRV, K^+^ and P^+^ levels in real time, suggesting that low electrolyte changes before and after HD alone may cause cardiac autonomic dysfunction.

## 1. Introduction

Heart rate variability (HRV) is determined by analyzing the changes in the heart, beat-to-beat [1]. HRV is an indicator of neurocardiac function and is generated by heart–brain interactions and dynamic autonomic nervous system (ANS) processes. Autonomic function of heart can be measured using non-invasive HRV, representative of the sympathetic and parasympathetic activities of the ANS in the sinus node. The emergence of interdependent regulatory systems operate over varying periods of time to promote adaptation to different environmental and psychological changes. It is an index of the regulation of visceral, cardiac and vascular tone, reflecting autonomic balance, blood pressure (BP), gas exchange and the diameter of blood vessels that regulate blood pressure [2].

End-stage kidney disease (ESKD) is associated with high rates of morbidity. Cardiovascular (CV) disease is the leading cause of death in patients receiving hemodialysis (HD) [3]. In those cases, autonomic neuropathy of hearts with a higher risk of arrhythmia may partially reflect the high CV mortality rate, in addition to diabetes mellitus, hypertension and hyperlipidemia [4,5]. HRV can evaluate CV autonomic neuropathy measuring of variations in heart rate [5] and has the advantage of using a noninvasive approach to measure the ANS activities [1]. An abnormal HRV primarily reflects the dysregulation between the sympathetic and parasympathetic nervous systems, and higher HRV values indicate greater variation between two consecutive beats, thus reflecting higher parasympathetic activity [6]. Frequency domain analysis of HRV has gained popularity with a broad application as a functional indicator of the ANS because it is noninvasive and easily accessible. Low HRV, which indicates impaired autonomic function, has been reported in patients undergoing HD [7].

Furthermore, reduced HRV has been associated with adverse CV events with higher mortality in those patients [3,4], while HD itself has been suggested to improve HRV [6,7,8]. However, data to predict HD efficiency and explain the association between electrolyte changes before and after HD and HRV in patients with ESKD are limited. Most studies analyzed 24 h electrocardiogram (ECG) recordings, which is a time-consuming and labor-intensive approach. Our study used a remote system for HRV recording and frequency/time domain analysis of HRV to determine whether short HRV measurements during HD predict electrolyte changes (K^+^ and P^+^) after HD in patients with ESKD.

## 2. Methods

### 2.1. Participants

All ESKD patients were screened for medication use and medical conditions. From October 2022 to December 2022, we recruited 75 patients (mean age, 62.1 ± 10.7 years) with ESKD, undergoing routine follow-ups at the nephrology outpatient clinic and HD center.

The study population consisted of ESKD patients on HD aged 18 years or older. Exclusion criteria included history of valvular or congenital heart disease, hepatic disease (known chronic liver disease), acute cardiovascular or cerebrovascular event within the preceding three months, major trauma or surgery within the preceding three months, hyperthyroidism, uncontrolled hypertension (HTN), diabetes mellitus (DM), pregnancy or treatment that might affect HRV parameters. Finally, 50 consecutive ESKD patients (20 men and 30 women, mean age: 66.3 ± 7.5 years) were enrolled and all patients were monitored to evaluate HRV and electrolyte monitoring during HD. These statistics were included in the analysis.

### 2.2. Ethical Statement

The study protocol and the informed consent requirement of individual patients were approved by the Ethics Committee of Kosin University Gospel Hospital (IRB No. 2022-06-012). We obtained written informed consent from all enrolled patients. This study was conducted in accordance with the principles of the latest version of the Declaration of Helsinki (2013).

### 2.3. Data Collection

All patients underwent electrocardiography (ECG) and chest X-rays. The cardiovascular status of all patients was assessed by the attending physicians using echocardiography and blood laboratory tests from the initial enrollment. From the database, the following information was collected: patient data including age, gender, body mass index; cardiovascular risk factors including hypertension (defined as having a systolic BP > 140 mm Hg and diastolic BP of greater than 90 mm Hg) and DM (defined as having a fasting plasma glucose level of greater than 126 mg/dL on 2 consecutive assessments or a level of HbA1c greater than 6.5%, or if the patient is currently undergoing treatment for DM); cardiovascular disease status including structural heart disease, congestive heart failure or a history of a disabling cerebral infarction or transient ischemic attack; and use of medication.

### 2.4. ECG-Monitoring Device

Hicardi^®^ (MEZOO Co., Ltd., Wonju-si, Gangwon-do, Republic of Korea) is an 8 g, 42 × 30 × 7 mm wearable ECG-monitoring patch-type device certified by the Ministry of Food and Drug Safety of Korea. The ECG signal was acquired at a 250 Hz sampling frequency and 14 bit resolution. This wearable device monitors and records single-lead ECGs, respiration, skin-surface temperature and activity. Data from the wearable patch were transferred through Bluetooth Low Energy to a mobile gateway, implemented as a portable smartphone application. All data were transmitted by the mobile gateway to a cloud-based monitoring server. A wearable patch was attached to the left sternal border after obtaining informed consent from all patients. The ECG signals and data were continuously recorded, and all ECG signals were reviewed by all cardiologists using a cloud-based monitoring system.

### 2.5. HRV Parameters

We performed HRV analysis in the frequency and time domains of wearable ECG recordings according to international guidelines [8]. An average of 225.7 ± 107.3 h of ECG data per patient was recorded during HD, and the HRV analysis was performed before and after HD according to the electrolyte (K+ and P+) level changes before and after HD. To assess HRV parameters, R–R intervals must be computed from wearable ECG recordings.

The following steps were performed to obtain the R–R interval time series.

The R peaks were detected using the geometric angle between two consecutive samples of the ECG signal [9]. The detected R peaks were then used to generate an R–R interval time series. To remove abnormal intervals caused by ectopic beats, arrhythmic events, missing data and noise, intervals <80% or >120% of the average of the last six intervals were excluded. Time domain parameters were calculated from the R–R interval time series.The R–R interval time series was resampled at 4 Hz using linear interpolation. The resulting series was detrended by eliminating linear trends [10]. After detrending, the power spectral density for the R–R interval time series was estimated using the Burg autoregressive model, where the order of the model was 33. In the time domain, we analyzed the R–R intervals, standard deviations of the R–R intervals, square root of the mean squared difference of successive R–R intervals and the percentage of adjacent N–N intervals that differed by more than 50 ms (NN50).

In the frequency domain analyses, we analyzed low frequency (LF, 0.04–0.15 Hz), which was an index of both sympathetic and parasympathetic activity, and high frequency (HF, 0.15–0.4 Hz), which represented the most efferent vagal (parasympathetic) activity to the sinus node. Very low frequency (VLF; 0.003–0.04 Hz) partially reflects thermoregulatory mechanisms, fluctuations in the activity of the renin–angiotensin system and the function of peripheral chemoreceptors. The LF/HF ratio, which reflects sympathovagal balance, was also calculated.

### 2.6. Assessment of Electrolytes

Electrolyte measurements were performed before and after HD in all patients. In patients who required assessments of electrolyte levels and other blood sample parameters including complete blood count and liver function tests, the measurements were performed several times individually.

### 2.7. Statistical Analysis

All continuous variables are expressed as mean ± standard deviation or median (25th and 75th interquartile range), depending on the distribution. For continuous data, the statistical significance of the differences was evaluated using Student’s *t*-test or the Mann–Whitney *U* test, depending on the data distribution. Categorical variables were presented as frequencies (percentages) and were analyzed using the chi-squared test. To determine whether any of the variables were independently related to HRV based on the electrolytes (K^+^ and P^+^) level changes before and after HD, a multivariate analysis of variables with *p*-values < 0.05 in the univariate analysis was performed using linear logistic regression analysis. All correlations were calculated using Spearman’s rank correlation test. All statistical analyses were conducted using SPSS statistical software (version 19.0; SPSS Inc., Chicago, IL, USA), and statistical significance was set at *p* < 0.05 (two-sided).

## 3. Results

A total of 50 patients (age, 66.3 ± 7.5 yr) with ESKD underwent continuous real-time ECG monitoring (225.7 ± 107.3 h) for HRV using a remote-monitoring system during HD (Figure 1). We compared HRV in relation to the electrolyte profile and changes after HD. HRV, ambulatory heart rate and respiratory rate were measured every 15 min in all patients during real-time ECG monitoring.

During monitoring, we simultaneously analyzed 2374 ECG data points for HRV, ambulatory heart rate and respiration rate for all patients. The baseline characteristics and echocardiographic parameters of all patients with ESKD are shown in Table 1.

Both time and frequency domain HRVs, except for nHF, were higher in patients with ESKD with lower K^+^ level changes than in those with higher K^+^ level changes after HD (Table 2). As shown in Table 3, both time and frequency domain HRVs were higher in patients with lower P^+^ level changes than in those with higher P^+^ level changes after HD.

In addition, the mean heart rates and incidence of arrhythmic events during HD were higher in patients with lower K^+^ and P^+^ level changes than in those with higher K^+^ and P^+^ level changes after HD (Table 2 and Table 3). As shown in Figure 2, the total arrhythmic burden (percentage) of the total heart rate was constantly higher during HD in patients with lower K^+^ and P^+^ level changes than in those with higher K^+^ and P^+^ level changes after HD.

Univariate analysis revealed that SDNN, min NN, nHF and QTc changes were associated with K^+^ changes after HD. In multivariate analysis, min NN and normalized HF were independent predictors of K^+^ changes after HD in patients with ESKD (Table 4(A)). Similarly, in the univariate analysis, SDNN, min NN, nLF, nHF and the LF/HF ratio were associated with P^+^ changes after HD. In the multivariate analysis, SDNN and min NN were independent predictors of P^+^ changes after HD in patients with ESKD (Table 4(B)).

The ROC curve in Figure 3A shows that a minimum NN of ≥474 ms and normalized HF of ≥0.0732 N.U. predicted effective HD (K^+^ change ≥ 0.5 after HD; for mininimum NN: AUC = 0.679; 95% confidence interval = 0.647–0.710, *p* < 0.001; for normalized HF: AUC = 0.603; 95% confidence interval = 0.571–0.636, *p* < 0.001). The ROC curve in Figure 3B shows that a minimum NN of ≥ 598 ms predicted effective HD (P^+^ change ≥ 2 after HD; for minimum NN: AUC = 0.799, 95% confidence interval = 0.781–0.818, *p* < 0.001).

## 4. Discussion

In our study, we evaluated heart rate and HRV simultaneously during HD in accordance with K^+^ and P^+^ level changes before and after HD in ESKD patients. The results demonstrate that poorly controlled K^+^ and P^+^ levels during HD are associated with higher HRV in patients with ESKD independently. This is consistently evidenced by the independent association between higher HRV, K+ and P+ levels in real time. These associations may be independent of HD efficiency. Our results therefore support the notion that cardiac autonomic dysfunction occurs in real time before electrolyte assessments routinely occur—when poorly controlled K^+^ and P^+^ levels after HD are measured and may play a role in predicting lower HD efficiency earlier in the course of ESKD.

HRV is a non-invasive measurement, representative of the ANS that reflects beat-to-beat variabilities in heart rates and has been successfully applied in chronic dialysis patients [5]. A previous study reported that the root mean square of successive differences between R–R intervals using a standard 12-lead ECG recording to access the time domain of HRV independently predicts mortality in ESKD patients [11].

Compared to HRV measurements by 24 h Holter ECG, a short-term measurement of the resting HRV consistently predicts the long-term outcome (more than 10 years) in ESKD patients [12]. Giordano et al. studied HRV changes during dialysis sessions in patients with ESKD. In their study, healthy controls had lower resting LF/HF ratios than did patients with ESKD. During dialysis therapy, patients with ESKD showed increasing LF/HF, which indicated an increase in sympathetic activity during ultrafiltration [13]. Previous studies reported that HRV measurement, which reflects various aspects of ANS activities, is a simple and useful tool to predict long-term mortality among patients undergoing HD [14,15]. However, data to predict HD efficiency and electrolyte changes after HD are limited. In our study, we compared HRV in relation to electrolyte changes (K^+^ and P^+^) after HD. ANS dysfunction has been reported to occur in more than 50% of patients with chronic HD [16]. Fatal arrhythmias, which can cause sudden cardiac death, may result from the autonomic innervation damage of the heart and vessels [17], induced by ischemia [18]. Both parts of ANS can be affected, with sympathetic dysfunction followed by parasympathetic impairment [6]. Decreased HRV is one of the earliest signs of ANS dysfunction, and previous studies (Framingham Heart Study) reported that HRV was associated with a risk of mortality inversely [19]. Reduced HRV was associated with the coronary vascular disease [20] and cerebral small vessel disease (CSVD) in DM patients independently [21]. Sympathetic activity increase and parasympathetic activity decrease inducing a state of alertness was associated with adaptation to stress [22]. Interestingly, metabolic syndrome, smoking habits and depressive disorder are associated with ANS dysfunction such as decreased parasympathetic activity and increased sympathetic activity [7,23].

Arrhythmias can occur frequently during HD, and previous study also reported ECG artifacts associated with HD mimicking arrhythmias [24]. Despite the challenges associated with the impact of fluid fluctuations and electrolyte changes on ECG parameters, further ECG monitoring study including HRV during HD has the potential to give practically useful and clinically meaningful information for diagnostic inference for risk stratification. In our study, patients with poorly controlled K^+^ and P^+^ levels during HD showed higher arrhythmic events including atrial/ventricular premature complexes, although their mean heart rate did not differ from those with well-controlled electrolyte levels after HD (Figure 2).

Although there was no previous study to assess the relationship in patients with ESKD according to electrolyte changes after HD, there have been conflicting reports that hypertension in the general population is associated with an increased decline in all HRV parameters [25,26]. A previous study reported that decreased activity of autonomic nervous function has also been preceded by the clinical hypertension development [27]. However, in our study, there was no significant difference of HRV according to BP and history of hypertension.

This is the first study to evaluate concurrent HRV and electrolyte changes (K^+^ and P^+^) in patients with ESKD after HD by using a remote monitoring system. Importantly, unlike previous studies [28,29], our study shows that all time and frequency domain measurements of HRV were associated with worsening electrolyte level changes after HD in detail. This may be explained by the fact that we used real-time remotely monitored ECG-derived HRV, which is more accurate compared to HRV derived from short-term ECG recordings. Additionally, to adjust for many potential confounders, objectively measured real-time respiration and physical activity were adjusted in the laboratory’s live studio using a remote monitoring system.

### Limitations

There are some limitations in our study. First, this study included a relatively small sample size, and was a single-center study derived from real world practice with inherent limitations in ESKD patients undergoing HD. However, in our study, there was sufficient for identifying significant associations between HRV and electrolyte level changes in ESKD patients using a real-time remote system for HRV monitoring. Our analyzed data revealed a clinically significant and practically important relationship between HRV and electrolyte changes (K+ and P+), particularly related to HD efficiency. Thus, the results of our study should be considered as more hypothesis generating, with future prospective studies being warranted to confirm our results. Also, the electrolyte indices were only measured twice (before and after each HD session). Sympathetic tone was not estimated via direct methods such as muscle sympathetic nerve activity or plasma catecholamine levels, which may have helped verify the activities of the sympathetic nervous system. However, these direct methods are invasive and less clinically useful, and their predictive values have not yet been established.

## 5. Conclusions

Higher HRV was independently associated with poorly controlled K^+^ and P^+^ level during HD in patients with ESKD. This is consistently evidenced by the independent association between higher HRV and K+ and P+ levels in real time, suggesting that low electrolyte changes before and after HD alone may cause cardiac autonomic dysfunction.

## Figures and Tables

**Figure 1 biomedicines-12-00474-f001:**
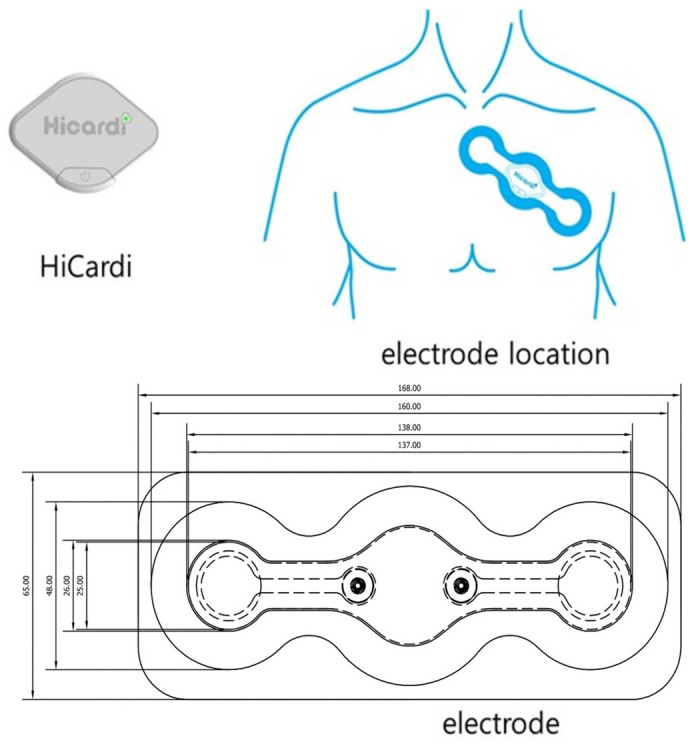
HiCardi device for HRV measurement.

**Figure 2 biomedicines-12-00474-f002:**
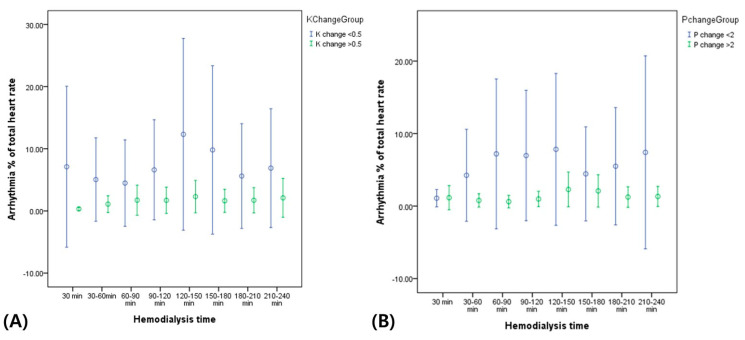
Arrhythmic events during HD according to (**A**) K^+^ change and (**B**) P^+^ change before and after HD.

**Figure 3 biomedicines-12-00474-f003:**
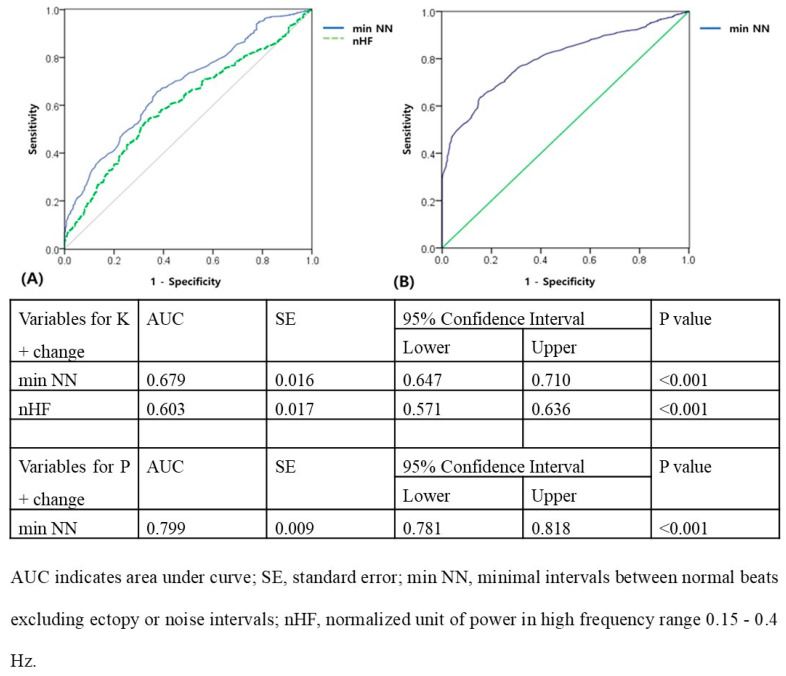
ROC curve for prediction of (**A**) K^+^ change and (**B**) P^+^ change before and after HD in patients with ESKD.

**Table 1 biomedicines-12-00474-t001:** Baseline demography and echocardiographic parameters in ESKD patients.

Variables	All ESKD Patients (*n* = 50)
Age (years)	62.2 ± 10.8
Gender (Male, %)	20 (40.0)
DM (%)	25 (50.0)
HTN (%)	44 (88.0)
CAD (%)	11 (22.0)
Stroke (%)	6 (12.0)
Medications	
Aspirin (%)	9 (18.0)
Statin (%)	28 (56.0)
Beta-blocker (%)	31 (62.0)
ACEi or ARB (%)	29 (58.0)
CCB (%)	25 (50.0)
Diuretics (%)	5 (10.0)
Hemodialysis parameters	
BFR (mL/min)	261.4 ± 10.7
DFR (mL/min)	510.0 ± 30.3
UFR (mL/min)	3072.0 ± 840.8
Kt/V	1.7 ± 0.2
URR	0.7 ± 0.05
Baseline Electrolytes	
Na^+^ (mEq/L)	136.4 ± 3.4
K^+^ (mEq/L)	4.9 ± 0.7
Ca^2+^ (mEq/L)	8.7 ± 0.6
P^+^ (mEq/L)	5.3 ± 1.4
CO^2^ (mEq/L)	21.7 ± 4.3
Echo parameters	
LVEF (%)	61.4 ± 11.2
LVIDs (mm)	33.2 ± 8.5
LVIDd (mm)	49.4 ± 7.8
IVSD (mm)	11.6 ± 2.3
LVPWD (mm)	9.9 ± 1.4
LAD (mm)	39.5 ± 7.1
E velocity (cm/sec)	0.8 ± 0.3
A velocity (cm/sec)	1.0 ± 0.0.3
E/A	0.9 ± 0.5
E/E’	17.1 ± 9.6

ESKD, end-stage kidney disease; DM, diabetes mellitus; HTN, hypertension; CAD, coronary artery disease; ACEi, angiotensin converting enzyme inhibitor; ARB, angiotensin receptor blocker; CCB, calcium channel blocker; BFR, blood flow rate; DFR, dialysate flow rate; UFR, ultrafiltration rate; Kt/V, number used to quantify hemodialysis and peritoneal dialysis treatment adequacy. [K—dialyzer clearance of urea, t—dialysis time, V—volume of distribution of urea]; URR, urea reduction rate; LVEF, left ventricular ejection fraction; LVIDs, left ventricular systolic diameter; LVIDd, left ventricular diastolic diameter; IVSD, interventricular septal diameter; LVPWD, left ventricular posterior wall diameter; LAD, left atrial diameter; E, the peak mitral flow velocity of the early rapid filling wave; A, peak velocity of the late filling wave due to atrial contraction; E’, early diastolic mitral annulus velocity.

**Table 2 biomedicines-12-00474-t002:** HRV and HR measurements in ESKD patients according to K^+^ level during HD.

HRV			
Time Domain	Lower K^+^ Level Change Group (<0.5 mEq/L)	Higher K^+^ Level Change Group (≥0.5 mEq/L)	*p*-Value
SDNN, ms	66.8 ± 39.1	43.6 ± 31.8	<0.001
RMSSD, ms	57.6 ± 7.7	42.0 ± 9.2	0.006
SDSD, ms	57.7 ± 7.7	42.1 ± 9.1	0.006
NN50, count	49.6 ± 7.5	30.4 ± 7.4	<0.001
pNN50, %	14.9 ± 1.9	9.6 ± 2.2	<0.001
Frequency domain			
nLF, N.U. × 10^6^	0.6 ± 0.2	0.5 ± 0.2	<0.001
nHF, N.U. × 10^6^	0.4 ± 0.2	0.5 ± 0.2	<0.001
LF/HF ratio	2.9 ± 0.2	2.3 ± 0.1	0.008
Variables			
Mean HR, beats/min	72.7 ± 12.1	69.3 ± 13.4	0.93
Arrhythmia (%)	7.2 ± 0.7	1.6 ± 0.9	<0.001
APC (total beats)	78.3 ± 14.5	1.4 ± 0.4	<0.001
VPC (total beats)	58.3 ± 12.8	23.9 ± 10.9	0.046

HRV, heart rate variability; HR, heart rate; ESKD, end-stage kidney disease; SDNN, standard deviation of NN intervals; RMSSD, root mean square of successive RR interval differences; SDSD, standard deviation of differences between adjacent NN intervals; NN50, number of NN intervals differing by more than 50 ms; pNN50, ratio of NN50; LF, power in low frequency range 0.04–0.15 Hz; HF, power in low frequency range 0.15–0.4 Hz; N.U., normalized unit; HR, heart rate; APC, atrial premature complex; VPC, ventricular premature complex. nHF, normalized unit of power in high frequency range 0.15–0.4 Hz. nLF, normalized unit of power in low frequency range 0.04–0.15 Hz.

**Table 3 biomedicines-12-00474-t003:** HRV and HR measures in ESKD patients according to P^+^ level during HD.

HRV			
Time Domain	Lower P^+^ Level Change Group (<2 mEq/L)	Higher P^+^ Level Change Group (≥2 mEq/L)	*p*-Value
SDNN, ms	62.1 ± 4.2	41.6 ± 1.1	<0.001
RMSSD, ms	80.8 ± 6.5	32.8 ± 1.4	<0.001
SDSD, ms	80.9 ± 6.5	32.8 ± 1.4	<0.001
NN50, count	53.3 ± 4.6	25.7 ± 1.4	<0.001
pNN50, %	16.1 ± 1.4	8.3 ± 0.4	<0.001
Frequency domain			
nLF, N.U. × 10^6^	0.5 ± 0.2	0.5 ± 0.2	<0.001
nHF, N.U. × 10^6^	0.5 ± 0.2	0.4 ± 0.2	<0.001
LF/HF ratio	2.1 ± 0.2	2.5 ± 0.8	0.011
Variables			
Mean HR, beats/min	82.0 ± 13.0	67.6 ± 11.6	<0.001
Arrhythmia (%)	5.6 ± 1.4	1.3 ± 0.3	<0.001
APC (total beats)	18.7 ± 6.8	8.0 ± 3.0	0.003
VPC (total beats)	81.5 ± 21.9	13.2 ± 3.4	<0.001

HRV, heart rate variability; HR, heart rate; ESKD, end-stage kidney disease; SDNN, standard deviation of NN intervals; RMSSD, root mean square of successive RR interval differences; SDSD, standard deviation of differences between adjacent NN intervals; NN50, number of NN intervals differing by more than 50 ms; pNN50, ratio of NN50; LF, power in low frequency range 0.04–0.15 Hz; HF, power in low frequency range 0.15–0.4 Hz; N.U., normalized unit. nHF, normalized unit of power in high frequency range 0.15–0.4 Hz. nLF, normalized unit of power in low frequency range 0.04–0.15 Hz. APC, atrial premature complex; VPC, ventricular premature complex.

**Table 4 biomedicines-12-00474-t004:** Univariate and multivariate Cox analyses for (A) K^+^ change and (B) P^+^ change in ESKD patients before and after hemodialysis.

(A)	Univariate Analysis		Multivariate Analysis	
Variable. *n* (%)	OR (95% CI)	*p*-Value	OR (95% CI)	*p*-Value
SDNN	0.996 (0.994–0.997)	<0.001		
minNN	1.004 (1.003–1.004)	<0.001	1.004 (1.003–1.005)	<0.001
nHF	4.933 (2.852–8.534)	<0.001	6.441 (3.610–11.489)	<0.001
QTc change	1.042 (1.002–1.084)	0.038		
(B)	Univariate Analysis		Multivariate Analysis	
Variable. *n* (%)	OR (95% CI)	*p*-Value	OR (95% CI)	*p*-Value
SDNN	0.996 (0.994–0.997)	<0.001	1.006 (1.004–1.006)	<0.001
minNN	1.006 (1.005–1.006)	<0.001	1.007 (1.006–1.008)	<0.001
nLF	3.039 (2.034–4.541)	<0.001		
nHF	0.336 (0.225–0.502)	<0.001		
LF/HF ratio	1.045 (1.010–1.081)	0.012		

OR, odd ratio; CI, confidence interval; SDNN, standard deviation of NN intervals; minNN, minimal intervals between normal beats excluding ectopy or noise intervals; nHF, normalized unit of power in high frequency range 0.15–0.4 Hz; nLF, normalized unit of power in low frequency range 0.04–0.15 Hz.

## Data Availability

Data availability statements are available after contact with the corresponding author; 67920@naver.com.

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
