# Peer review of "Hemodialysis Efficiency Predictor in End-Stage Kidney Disease Using Real-Time Heart Rate Variability"

_biomedicines, 2024, doi:10.3390/biomedicines12030474_

Round 1

Reviewer 1 Report

Comments and Suggestions for Authors

The paper describes a tool to predict the efficiency of hemodialysis (HD) in end stage kidney disease patients based on the real-time analysis of heart rate variability that is calculated from ECG recordings. The variability relates to the change of electrolyte levels in patients during HD. The variability was analyzed both in time and frequency domain.

For a reader with no specific knowledge on this field like me the introduction does not work well, and as a consequence neither does the introduction part within the abstract. For instance,

- what is the role of the electrolyte levels in the story? I guess it’s not about the absolute levels, but about the change of the levels during hemodialysis, but I’m not sure from the introduction.

- I guess higher HRV levels are problematic, but page 2 first section suggests that also low HRV levels are too

These comments are also presented to the authors.

Comments on the Quality of English Language

The English language is good, but can be improved.

In abstract: the first time using the acronym HD should be given as hemodialysis (HD)

In introduction:

- line 3: where does ‘which’ refer to: ‘cardiac autonomic function’ or ‘HRV’?

- HRV is described differently in different sentences, which does not help understanding the role of HRV

- last sentence: too many ‘;’’

Author Response

Biomedicines

Manuscript title: Hemodialysis Efficiency Predictor in End Stage Kidney Disease Using Real-Time Heart Rate Variability

Manuscript ID: biomedicines-2847733

Authors: Ho Sik Shin (Corresponding author)

We appreciate reviewers for their efforts in evaluating our commentary. We also appreciate the reviewers’ very helpful and precious comments, which helped a lot for us to improve our manuscript. The reviewer’s specific comments are cited in italics and highlighted in blue and our detailed responses are as followings.

Point-by-point Responses

COMMENTS TO AUTHOR:

Author's Reply to the Review Report (Reviewer 1)

Please provide a point-by-point response to the reviewer’s comments and either enter it in the box below or upload it as a Word/PDF file. Please write down "Please see the attachment." in the box if you only upload an attachment. An example can be found here.

Comments and Suggestions for Authors

The paper describes a tool to predict the efficiency of hemodialysis (HD) in end stage kidney disease patients based on the real-time analysis of heart rate variability that is calculated from ECG recordings. The variability relates to the change of electrolyte levels in patients during HD. The variability was analyzed both in time and frequency domain.

For a reader with no specific knowledge on this field like me the introduction does not work well, and as a consequence neither does the introduction part within the abstract. For instance,

- what is the role of the electrolyte levels in the story? I guess it’s not about the absolute levels, but about the change of the levels during hemodialysis, but I’m not sure from the introduction.

: Thank you for comments. Actually, we discussed about ANS dysfuction in ESKD patients and about the topics of our study with nephrologist. And they (nephrologists) wanted to find out good predictors for hemodialysis efficiency. because, even though there are lots of parameters, such as Kt/v, ultrafiltration rate etc., there is no good parameter to predict hemodialysis efficiency for all ESKD patients undergoing hemodialysis routinely (every 2 days). In real world practice, nephologist wants to explain to ESKD patients how hemodialysis is going on such as dialysis well today or not. And they mentioned that the most important electrolyte changes for ESKD to predict hemodialysis efficiency are K+ and P+ changes before/after hemodialysis, especially P+ changes. So, we analyzed data according to K+ and P+ changes.

- I guess higher HRV levels are problematic, but page 2 first section suggests that also low HRV levels are too

: Thank you for comments. as we know, higher HRV levels are problematic. However, too lower HRV levels also can be problematic. Because too lower ANS activity can be problematics.

So we mentioned about those (ref.; PLoS ONE 2018, 13, e0195166)

These comments are also presented to the authors.

Comments on the Quality of English Language

The English language is good, but can be improved.

In abstract: the first time using the acronym HD should be given as hemodialysis (HD)

In introduction:

- line 3: where does ‘which’ refer to: ‘cardiac autonomic function’ or ‘HRV’?

: Thank you for comments. in introduction line 3, ‘which’ refer to ‘HRV’

- HRV is described differently in different sentences, which does not help understanding the role of HRV

: Thank you for comments. as reviewer’s comments, we revised as follow in page 2, line 3:

“HRV is an indicator of neurocardiac function and is generated by heart-brain interac-tions and dynamic autonomic nervous system (ANS) processes. Cardiac autonomic function can be assessed non-invasively by using HRV, which reflects the interaction of the sympathetic and parasympathetic parts of the ANS in the sinoatrial node.”

- last sentence: too many ‘;’’

:Thank you for comments. as reviewer’s comments, we deleted those.

Reviewer 2 Report

Comments and Suggestions for Authors

The study presents a comprehensive exploration of the relationship between autonomic neuropathy, heart rate variability (HRV), and electrolyte levels in patients with end-stage kidney disease (ESKD) undergoing hemodialysis (HD) which adds a valuable contribution to the limited data available in the field. However few questions need to be addressed.

1. Can you provide additional details on the selection criteria for the 50 patients with end-stage kidney disease (ESKD) included in the study?

2. Were there specific characteristics or comorbidities considered for inclusion or exclusion?

3.How was the continuous real-time ECG monitoring system implemented during hemodialysis (HD)? Were there any technical challenges or considerations that might have influenced the accuracy of the recorded data?

4. In the comparison of heart rate variability (HRV) based on electrolyte levels, were there any confounding factors or variables controlled for, such as medication use or pre-existing cardiac conditions, that might have influenced the results?

5. How generalizable are these findings to the broader population of patients with ESKD undergoing hemodialysis? 

6. Given the recruitment period from October 2022 to December 2022, were there any seasonal or environmental factors that might have influenced the results? 

Author Response

Biomedicines

Manuscript title: Hemodialysis Efficiency Predictor in End Stage Kidney Disease Using Real-Time Heart Rate Variability

Manuscript ID: biomedicines-2847733

Authors: Ho Sik Shin (Corresponding author)

We appreciate reviewers for their efforts in evaluating our commentary. We also appreciate the reviewers’ very helpful and precious comments, which helped a lot for us to improve our manuscript. The reviewer’s specific comments are cited in italics and highlighted in blue and our detailed responses are as followings.

Point-by-point Responses

COMMENTS TO AUTHOR:

Author's Reply to the Review Report (Reviewer 2)

Please provide a point-by-point response to the reviewer’s comments and either enter it in the box below or upload it as a Word/PDF file. Please write down "Please see the attachment." in the box if you only upload an attachment. An example can be found here.

Comments and Suggestions for Authors

The study presents a comprehensive exploration of the relationship between autonomic neuropathy, heart rate variability (HRV), and electrolyte levels in patients with end-stage kidney disease (ESKD) undergoing hemodialysis (HD) which adds a valuable contribution to the limited data available in the field. However few questions need to be addressed.

  1. Can you provide additional details on the selection criteria for the 50 patients with end-stage kidney disease (ESKD) included in the study?

: Thank you for comments. Actually, at the beginning of our study, we discussed about the topics with nephrologist. They wanted to find out good predictors for hemodialysis efficiency. because, even though there are lots of parameters, such as Kt/v, ultrafiltration rate etc., there is no good parameter to predict hemodialysis efficiency for all ESKD patients undergoing hemodialysis routinely (every 2 days). In real world practice, nephologist wants to explain to ESKD patients how hemodialysis is going on such as dialysis well today or not. So in our study, we enrolled all ESKD aged >18 years who underwent HD. The main exclusion criteria were pregnancy, neurological disease, heart failure, chronic liver failure, uncontrolled diabetes mellitus (DM), thyroid disorder, and treatment that could influence HRV parameters.

  1. Were there specific characteristics or comorbidities considered for inclusion or exclusion?

: Thank you for comments. we included all ESKD patients undergoing hemodialysis routinely , irrespective of co-morbidities including hypertension, coronary heart disease etc. and we compared co-morbidities according to electrolyte changes (K+ and P+) before/after hemodialysis and there was no significant differences of co-morbidities in both groups.

3.How was the continuous real-time ECG monitoring system implemented during hemodialysis (HD)? Were there any technical challenges or considerations that might have influenced the accuracy of the recorded data?

: Thank you for comments. we used patch type (Hicardi®) real-time ECG monitoring system and implemented those patch type ECG monitoring system in all patients who agreed to join before hemodialysis and removed those after hemodialysis. And Patch type real time ECG monitoring system is very user-friendly device that can be easily implemented. And during the monitoring, the nurses and technicians were observing patients and monitoring system at another room near Nurse station in Hemodialysis room. Actually, 2 patients had unclear signals of ECG monitoring. So we did change those (Hicardi®) 2 times respectively before hemodialysis associated with contact issues. And we recommended to all patients to lay down during hemodialysis – not to stand up.

  1. In the comparison of heart rate variability (HRV) based on electrolyte levels, were there any confounding factors or variables controlled for, such as medication use or pre-existing cardiac conditions, that might have influenced the results?

: Thank you for comments. yes, we agreed the reviewer’s comments and also considered that those variables such as medications, pre-existing cardiac conditions can be confounding factors. So we compared those according to electrolyte changes. However, there was no significant differences of those comorbidities and medications in both groups. And there was no significant association between results and those parameters using Cox analysis (univariate & multivariate analysis)

  1. How generalizable are these findings to the broader population of patients with ESKD undergoing hemodialysis?

: Thank you for comments. As we mentioned above, actually, at the beginning of our study, we discussed about the topics with nephrologist. They wanted to find out good predictors for hemodialysis efficiency. because, even though there are lots of parameters, such as Kt/v, ultrafiltration rate etc., there is no good parameter to predict hemodialysis efficiency for ESKD patients undergoing hemodialysis routinely (every 2 days). In real world practice, nephologist wants to explain to ESKD patients how hemodialysis is going on such as dialysis well today or not. So, we reviewed some data published previously and we hypothesized if the HRV as representative of ANS activity can be good predictor for Hemodialysis.

And, to generalize these parameters for boarderline ESKD population, we think more large volume-prospective study to be needed. Thank you again for very impotant point.

  1. Given the recruitment period from October 2022 to December 2022, were there any seasonal or environmental factors that might have influenced the results?

: Thank you for comments. yes, we agreed the reviewer’s comments. however, we are not sure about time and environmental factors. In our study, to decrease those factors, we selected hemodialysis room without noise and all patients who were enrolled in our study, underwent in the morning. We also think there will be needed more large volume prosective study to clarify about those factors.

Submission Date

14 January 2024

Date of this review

24 Jan 2024 17:38:05

Thank you so much and hope our manuscript can be accepted in your premier journal.

Hope to hear good news from the editor.

Thank you.

Sincerely,

Sung Il Im/Ho Sik Shin
